# Genome Wide Association Study Uncovers the QTLome for Osmotic Adjustment and Related Drought Adaptive Traits in Durum Wheat

**DOI:** 10.3390/genes13020293

**Published:** 2022-02-02

**Authors:** Giuseppe Emanuele Condorelli, Maria Newcomb, Eder Licieri Groli, Marco Maccaferri, Cristian Forestan, Ebrahim Babaeian, Markus Tuller, Jeffrey Westcott White, Rick Ward, Todd Mockler, Nadia Shakoor, Roberto Tuberosa

**Affiliations:** 1Department of Agricultural and Food Sciences, University of Bologna, 40127 Bologna, Italy; condorelli87@gmail.com (G.E.C.); e.groli@sisonweb.com (E.L.G.); marco.maccaferri@unibo.it (M.M.); cristian.forestan@unibo.it (C.F.); 2School of Plant Sciences, College of Agriculture and Life Sciences, Maricopa Agricultural Center, University of Arizona, Maricopa, AZ 85138, USA; maria.newcomb2@usda.gov (M.N.); rickw.ward@gmail.com (R.W.); 3Department of Environmental Science, University of Arizona, Tucson, AZ 85721, USA; ebabaeian@arizona.edu (E.B.); mtuller@cals.arizona.edu (M.T.); 4US Arid Land Agricultural Research Center, USDA-ARS, Maricopa, AZ 85138, USA; jeff.white.az@gmail.com; 5Danforth Foundation, St. Louis, MO 63132, USA; tmockler@danforthcenter.org (T.M.); nshakoor@danforthcenter.org (N.S.)

**Keywords:** drought, durum wheat, osmotic adjustment, QTL

## Abstract

Osmotic adjustment (OA) is a major component of drought resistance in crops. The genetic basis of OA in wheat and other crops remains largely unknown. In this study, 248 field-grown durum wheat elite accessions grown under well-watered conditions, underwent a progressively severe drought treatment started at heading. Leaf samples were collected at heading and 17 days later. The following traits were considered: flowering time (FT), leaf relative water content (RWC), osmotic potential (ψs), OA, chlorophyll content (SPAD), and leaf rolling (LR). The high variability (3.89-fold) in OA among drought-stressed accessions resulted in high repeatability of the trait (h^2^ = 72.3%). Notably, a high positive correlation (r = 0.78) between OA and RWC was found under severe drought conditions. A genome-wide association study (GWAS) revealed 15 significant QTLs (Quantitative Trait Loci) for OA (global R^2^ = 63.6%), as well as eight major QTL hotspots/clusters on chromosome arms 1BL, 2BL, 4AL, 5AL, 6AL, 6BL, and 7BS, where a higher OA capacity was positively associated with RWC and/or SPAD, and negatively with LR, indicating a beneficial effect of OA on the water status of the plant. The comparative analysis with the results of 15 previous field trials conducted under varying water regimes showed concurrent effects of five OA QTL cluster hotspots on normalized difference vegetation index (NDVI), thousand-kernel weight (TKW), and/or grain yield (GY). Gene content analysis of the cluster regions revealed the presence of several candidate genes, including bidirectional sugar transporter SWEET, rhomboid-like protein, and S-adenosyl-L-methionine-dependent methyltransferases superfamily protein, as well as DREB1. Our results support OA as a valuable proxy for marker-assisted selection (MAS) aimed at enhancing drought resistance in wheat.

## 1. Introduction

Drought is one of the most devastating abiotic stressors limiting crop yield, adaptability, and quality [1,2]. Recent global climate models predict a consistent rainfall reduction in temperate drylands [3,4,5], hence destabilizing food systems and global food security [6]. The plant reaction to drought is mediated by complex molecular systems linked to the transcriptome [7,8,9,10], as well as hormone signaling and metabolism [11,12,13]. In particular, drought is the major abiotic stress curtailing yield and lowering quality [14,15] in durum wheat (*Triticum turgidum* ssp. *durum*; 2n = 28, AABB), the most cultivated wheat in the Mediterranean regions [16], whose genome sequence was recently assembled *de novo* [17]. Among the strategies adopted by plants to withstand water scarcity [18,19], osmotic adjustment (OA) plays a major role in enhancing drought resistance through an active accumulation of solutes in response to a water potential reduction, thereby preserving cellular turgor [18,20,21,22,23,24,25,26,27]. Active OA maintains relative water content at low leaf water potential in order to sustain plant growth without impairing normal cellular functions [28]. Plants accumulate low-molecular weight organic solutes, such as soluble sugars [24,29] and proline [27,30,31,32], both of which increase under water stress, hence enhancing OA and contributing to maintain photosynthesis, as well as stomatal conductance, at lower water potentials. To date, the dissection of the genetic basis of OA has received limited attention, mainly due to the difficulty in measuring this trait in more than a limited number of accessions [33], an essential prerequisite to properly map and characterize the effects of the QTLs [34,35,36] underscoring OA variability. In cereals, the dissection of the OA QTLome has been attempted in rice [37,38] and barley [39,40] based on the evaluation of biparental recombinant inbred lines (RIL) populations, hence surveying only a limited amount of genetic variability as compared to that surveyed in GWAS studies. Herein, we report the results of the first large-scale genetic dissection of the OA QTLome in wheat via GWAS based on the field evaluation of 248 durum wheat elite accessions grown under conditions of progressively increasing drought and previously tested for grain yield in 15 field trials carried out under a broad range of water regimes in Mediterranean countries [41]. Three major QTL clusters were identified, where OA was unrelated to flowering time while being positively associated with the water status of the plant and grain yield as reported in [41], supporting the beneficial role of OA in enhancing drought resistance, most likely through an avoidance strategy. A comparative analysis with the sequence information available for these regions in durum [17] and bread wheat [42] revealed a number of putative candidate genes.

## 2. Materials and Methods

### 2.1. Plant Material and Field Management

For this study, 248 durum wheat elite accessions (Durum Panel) were chosen at the University of Bologna (Appendix A). Most (189) of these accessions were originally assembled by [43] to represent a large portion of the genetic diversity (Appendix A) present in the major improved durum wheat gene pools adapted to Mediterranean environments. The field trial was conducted at the University of Arizona Maricopa Agricultural Center (33.070 °N, 111.974 °W, elevation 360 m) on a Casa Grande Soil (fine-loamy, mixed, superactive, hyperthermic Typic Natrargids) (Appendix A). The Durum Panel was planted on 28 November 2017 according to a row-column experimental design with two replicates. Each accession was evaluated in two-row plots (3.5-m long, 0.76-m row spacing) with an average plant density of 22 plants/m^2^. Orita and Tiburon, both representing the Arizona’s “Desert-Durum”^®^ wheat, were chosen as border plots. Before planting, granular nitrogen at 112 kg ha^−1^ was incorporated into the soil. Sprinkler irrigation was used to germinate seeds and establish the crop, followed by subsurface drip irrigation matching evapotranspiration for optimal plant growth, once or twice a week as needed. The pressurized subsurface drip irrigation system was installed before planting when one dripline with emitters spaced every 0.30 m was buried at ~0.10 m depth along each seed row. The final irrigation event was on 11 March 11 2018 (i.e., 103 days after sowing, DAS), when ~50% of the accessions had flag leaf sheaths opened (i.e., at Zadoks growth stage 47) [44]. From here on, the whole experiment was subjected to a progressive water deficit until 2–3 April 2018, when plants at the anthesis halfway stage (Zadoks growth stage 65, on average) were harvested to measure total above-ground biomass.

### 2.2. Meteorological Data and Soil Moisture Monitoring 

Daily and hourly meteorological reports for the growing season were obtained from the Arizona Meteorological Network [45]. In addition, high temporal resolution meteorological data, particularly air temperature, relative humidity, and photosynthetic photon flux density (PPFD), for the experimental site were recorded at 5-s intervals with an automated weather station (Clima Sensor US, Adolf Thies GmbH & Co. KG, Göttingen, Germany) and a quantum sensor (SQ-214, Apogee Instruments, Inc., Logan, UT, USA). These data were made available by the TERRA Phenotyping Reference Platform [46]. Vapor pressure deficit (VPD) was calculated as the difference between the saturation and actual vapor pressure [47]. The soil volumetric water content (VWC) was monitored in and between seed rows with time-domain reflectometry (TDR) sensors (True TDR-315, Acclima, Inc., Meridian, ID, USA) installed at three locations within the experiment at 1, 10, and 50 cm depths at each location. Additional soil sensors were installed between rows at 15 cm depth to measure the soil matric potential (Tensiomark, ecoTech Umwelt-Meßsysteme GmbH, Bonn, Germany). All soil sensors recorded data at 15-min intervals throughout the entire growing season. Based on the characterization of the soil hydraulic and physical properties of the experimental site under the TERRA-REF project, the volumetric water contents corresponding to the permanent wilting point (θPWP) and the field capacity (θFC) at 10–15 cm depth were approximately 0.110 and 0.282 m^3^/m^3^, respectively. The VWC dynamics at the three measurement locations, for the entire growing season, are depicted in Appendix A. The Durum Panel accessions were monitored regularly for above-ground diseases and pests, which remained below threshold levels, hence not requiring control treatments, while growing degree days (GDD) were monitored until harvest (3-4 April 2018) at 125 days after sowing (DAS) (Appendix A). Growth stages of each accession were defined based on the basis of the Zadoks scale [44] at 92, 93, 98, 101, 111, and 118 DAS, and flowering time (FT) was recorded when more than 50% of ears in the plot had flowered (anthesis half-way). Plants were harvested 125 days (DAS) to allow for planting the next phenotyping experiment; therefore, biomass data indicate the status at a point in time rather than direct estimates of final yields.

### 2.3. Evaluation of RWC, ψs, OA, LR, Leaf Chlorophyll Content (SPAD), and Biomass

The entire Durum Panel was evaluated for leaf relative water content (RWC) and osmotic potential (ψs) in well-watered (12 March 2018, 104 DAS) and severe drought (27 March 2018, 119 DAS) conditions. At the first sampling (fully-irrigated conditions) awns were visible on approximately 50% of accessions, while the second sampling was carried out under severe drought conditions when most accessions were at early grain-filling (Appendix A). Fully expanded flag leaves of eight different plants were sampled in each plot (experimental unit) at dawn from 6:00 to 7:00 a.m. Leaves were immediately placed in sealed plastic bags, stored in portable coolers (4 °C) to minimize water loss due to evaporation, and transported to the lab where leaves were removed from the bags. After cutting the leaf tips (5 cm), the remaining leaf portion (average length 15 cm) was cut in the middle to obtain two homogeneous pieces of similar weight, then mixed and stored in Falcon (50 ml) conical centrifuge tubes. One batch was used to measure OA following the “Rehydration method” described in Reference [23]. Leaves were rehydrated for 4 h in distilled water to reach full turgor, then dried, and stored in a freezer (−20 °C). After thawing, the cell sap was collected using a garlic press, and 10 µl were transferred onto a paper sample disc covering the sampling cuvette of a vapor pressure osmometer (Wescor 5520, Logan, UT, US), previously calibrated using the 290, 1000, and 100 mmol kg^−1^ standards. After each measurement, the osmometer cuvette was rinsed using deionized water. Finally, the resulting osmolality (mosmol kg^−1^) was converted to osmolarity (MPa) using the following formula: ψs (MPa) = −c (mosmol kg^−1^) × 2.58 × 10^−3^ [48], and OA was measured as the difference between the ψs at full turgor in control and in water-stressed conditions: ψs (control)—ψs (water stress). The other batch was used to measure RWC. Fresh leaves were weighed (FW) then submerged in distilled water in the Falcon tubes and stored at 4 °C for rehydration (10 h). Rehydrated leaves were wiped thoroughly with blotting paper and weighed (turgid weight: TW). Then, leaves were oven-dried at 65 °C for three days prior to measuring the dry weight (DW). In the end, RWC values were computed as follows: [(FW-DW)/(TW-DW)] × 100 [49] (Figure 1). Leaf rolling (LR) was visually estimated at midday (112 DAS) with a 0 (no leaf rolling) to 9 (all leaves severely rolled) score when the majority of the accessions showed a LR > 5. Finally, the chlorophyll content was assessed (114 DAP) based on Soil-Plant Analysis Development (SPAD) estimates obtained with a non-destructive chlorophyll meter SPAD-502Plus (Konica Minolta Sensing, Inc., Osaka, Japan) as an indicator of leaf chlorophyll content and nitrogen (N) status. At the end of the field trial, plants within the entire two-row plots were cut on 3–4 April 2018 with mechanical harvester (Carter Mfg. Co. equipment, Donalsonville, GA, US), while subsamples of 2–3 plants were collected to evaluate moisture content in order to estimate dry biomass. 

### 2.4. Statistical Analysis

The *lme4* package (*r-project*) and custom R scripts were used to conduct a spatial adjustment analysis considering row and column as random effects, as well as a moving mean of specific size. *R-project* was used to calculate repeatability values (*h^2^*) and Pearson’s correlation *r* coefficients among traits. *Minitab 18* software [50] was used to calculate the global percentage of phenotypic variation (global QTL model, *R^2^*%) explained by all QTLs for each trait. 

### 2.5. SNP Genotyping, Population Structure, and GWAS Model

Durum panel genomic DNA was extracted using the NucleoSpin® 8/96 Plant II Core Kit from Macherey Nagel and sent for SNP genotyping to [51]. The Illumina iSelect 90K wheat SNP assay [52] was used, and genotype calls were acquired as reported in [53]. Markers were assigned on the basis of the tetraploid wheat consensus map reported in [54]. Haploview 4.2 software [55] was used to calculate Linkage Disequilibrium (LD) decay among markers for the A and B genomes, and only Single Nucleotide Polymorphisms (SNPs) with minor allele frequency (MAF) > 0.05 were considered. LD decay pattern based on the consensus genetic distances was inspected considering squared allele frequency correlation (r^2^) estimates from all pairwise comparisons among intra-chromosomal SNPs in TASSEL (Trait Analysis by aSSociation, Evolution, and Linkage) *5.2.37*. The Hill and Weir formula [56] was used in R-project to define the confidence interval (CI) for QTLs in accordance with the curve fit and the distance at which LD decays below r^2^ = 0.3 [57]. Haploview 4.2 tagger function set to r^2^ < 1.0 was used to calculate a kinship matrix (K) of genetic relationships among individual accessions of the Durum Panel with all non-redundant 7,723 SNPs. Kinship based on Identity-by-State (IBS) among accessions was calculated in TASSEL 5.2.37. In addition, a subset of non-redundant 2,382 SNP markers (*r^2^* < 0.5) was used to evaluate the population structure (Q) in STRUCTURE 2.3.4. software [58] using the corresponding tagger function in Haploview 4.2 software [55]. Numbers of hypothetical subpopulations ranging from k = 2 to 10 were assessed using 50,000 burn-in iterations, followed by 100,000 recorded Markov-Chain iterations, in five independent runs for each k in order to estimate the sampling variance (robustness) of population structure inference. Then, the rate of change in the logarithm of the probability of likelihood (LnP(D)) value between successive k values was considered (Δk statistics) [58], together with the rate of variation (decline) in number of accessions clearly attributed to subpopulations (accessions with Q membership’s coefficient ≥ 0.5). Finally, the level of differentiation among subpopulations was measured using the Fixation Index (Fst) among all possible population pairwise combinations [59]. Subsequently, 17,721 SNPs with MAF > 0.05, imputed with LinkImpute (LDkNNi) [60,61], were used for GWAS-Mixed Linear Model [MLM; [62,63] in TASSEL. MLM was specified as follows: y = Xβ + Zu + e [64], where y is the phenotype value, β is the fixed effect due to the marker, and u is a vector of random effects not accounted for by the markers; *X* and *Z* are incidence matrices that related y to β and u, while e is the unobserved vector of random residual. In this study, both Kinship matrix (K) and Structure Population (Q) were included as random effects in the model (MLM-Q+K), while flowering time was included as a covariate considering GWAS QQ-plot results (Appendix A). Then, GWAS *p-*values and r^2^ effects were analyzed, and QTL significance was determined as follows: “highly significant” refers to *p-*value < 0.0001 and “significant” refers to *p-*value < 0.001. The QTL confidence interval (CI, in cM) was measured on the basis of the average genetic distance at which LD decayed below r^2^ of 0.3 [56], a threshold frequently adopted in GWAS [54,57,65]. Considering a LD of r^2^ = 0.3, the corresponding inter-marker genetic distance was 3.0 cM [57], and the CI of ± 3.0 cM based on map positions of QTL tag-SNPs was chosen. Finally, Minitab 18 software [50] was used to calculate the proportion of variance for phenotypic traits explained by selected SNPs. 

### 2.6. Identification of Candidate Genes

The physical position of each QTL was determined by the position of the flanking SNP markers after their alignment on the *Triticum turgidum* ssp. *durum* reference genome of (cv. Svevo) [66]. Genes within the confidence intervals associated with the eight main QTL hotspots were retrieved from the EnsemblPlants database [67], together with their functional annotation and the amino acid sequences of putative proteins. Gene Ontology (GO) term enrichment was determined by comparing the genes included in each QTL to the number of genes annotated in each GO term with g:Profiler web software [68]. Statistical significance of terms for genes in the physical intervals was assessed using the hypergeometric statistic for every term and the g:SCS correction method for multiple testing. Durum wheat GO annotation was retrieved from the Ensembl plant genome database. To identify the most important metabolic pathways associated to eight QTLs, genes within cluster intervals were aligned to KEGG (Kyoto Encyclopedia of Genes and Genomes) database using Reference [69]. Genes annotated within the intervals were compared with their orthologs from *Triticum aestivum* (cv. Chinese Spring; IWGSC RefSeq v1.0) [70]. Identification of candidate genes was further supported by a knowledge network (proteins, biological pathways, phenotypes, and publications) created using the KnetMiner program, using the bread wheat orthologs [71], and by the analysis of temporal and spatial gene expression at the Wheat Expression Browser and ePLANT databases [72], as of September 2021. 

## 3. Results

### 3.1. Population Structure

The Durum Panel showed a clear population genetic structure with an optimal number of eight (k = 8) subpopulations on the basis of pairwise comparisons among and within subgroups with 155 accessions (62.5%) clearly grouped into one of the eight main gene pools at a Q membership coefficient ≥ 0.5, while the remaining 93 were considered as admixed. The Fixation Index (Fst) and Neighbor Joining tree [73] highlighted a high genetic diversity between the old Italian accessions (S1) and the modern French, North American, Canadian and Australian cultivars (S8), while a considerable admixture among subgroups characterized the ICARDA, CIMMYT, and Italian groups. Subgroups details are reported in Appendix A.

### 3.2. Phenotypic Analysis

Osmotic potential in well-watered (control) conditions (ψs-c; h^2^ = 0.57) ranged from −1.44 to −0.74 MPa, with an average of −1.13 MPa, while, in water-stressed conditions, (ψs-s; h^2^ = 0.58) ranged from −2.63 to −1.56 MPa, with an average of −2.00 MPa. The difference between osmotic potential measured at full turgor in well-watered (control: ψs-c) and in water stressed (ψs-s) conditions was considered to compute OA (h^2^ = 0.72), which showed a normal distribution and ranged from 0.38 to 1.48 MPa, with an average of 0.95 MPa. Appendix A reports flowering time distribution, while Appendix A report box plots and histogram distributions, for OA, ψs-c and ψs-s, RWC-c and RWC-s, LR, and SPAD. RWC-c (h^2^ = 0.29) ranged from 89.9 to 101.3%, with an average of 95.7%, while RWC-s (h^2^ = 0.78) ranged from 45.2 to 76.9%, with an average of 62.2% (Appendix A). Leaf rolling (LR; h^2^ = 0.84) at 112 DAP ranged from 2.86 to 9.60, with an average of 6.13 (Table 1), while leaf chlorophyll content (SPAD; h^2^ = 0.76) at 114 DAP ranged from 31.9 to 48.8, with an average of 42.0. The Pearson’s correlation coefficient was positive between OA and RWC-s (r = 0.78), while a negative association was found between OA and LR (r = −0.25), RWC-s and ψs-s (r = −0.49), and RWC-s and LR (r = −0.30) (Table 2, Figure 2).

### 3.3. Genetic Analysis 

The rate of linkage disequilibrium (LD) decay of the 248 durum wheat elite accessions of the Durum Panel is reported in Figure 3. The average QTL confidence interval (CI) was determined on the basis of the average genetic distance at which LD decayed below r^2^ of 0.3 multiplied by 2, corresponding to 2.12 cM (CI = ± 1.06 cM from the QTL tagSNP). Fifteen flowering time QTLs were identified and are reported in Appendix A. Major QTLs for flowering time included those on chromosome arms 2AS (*QFT.ubo-2A.1* and *QFT.ubo-2A.2*), on 4AS (*QFT.ubo-2A.1*) and 6BL (*QFT.ubo-6B.1*). Among others, *Ppd-A1* was clearly identified by *QFT.ubo-2A.2* = IWA2526. Using FT as covariate, GWAS analysis (MLM-Q+K) identified 70 significant QTLs (log *p-*value > 3.00) for ψs-c, ψs-s, OA, RWC-s, LR, and/or SPAD, organized into QTL clusters. A larger portion of ψs QTLs were detected under drought (62.5%) as compared to well-watered conditions (37.5%). In particular, two major ψs-s QTLs were observed on chromosomes 1B (*Qψsc.ubo-1B.2*) and 6A (*Qψsc.ubo-6A.1*), with a log *p-*value of 4.68 and 6.04, and with R^2^ of 5.84 and 7.88%, respectively. A total of 15 OA QTLs were mapped on chromosome arms 1AL, 1BL, 2AS, 2AL, 2BL, 4AL, 4BS, 5AL, 6AL, 6BL, and 7BS, with the three major ones being those on chromosomes 2B (*QOA.ubo-2B.2*) and 6A (*QOA.ubo-6A.1*and *QOA.ubo-6A.2*) with a log *p-*value of 4.13, 4.01, and 4.45, and R^2^ of 4.37, 4.23, and 4.78%, respectively (Table 3). Adopting flowering time as a covariate effectively removed the effects associated to FT on OA, except for *QFT.ubo-2A.2* = *QOA.ubo-2A.1* and *QFT.ubo-6B.1* = *QOA.ubo-6B.1*. Nine RWC-s loci were mapped on five chromosome arms (1BS, 2AS, 4AL, 6AL, and 6BL), and two major QTLs were observed on 4AL (*QRWCs.ubo-4A.2* and *QRWCs.ubo-4A.3*), with a log *p-*value of 4.83 and 4.27, and R^2^ of 3.95 and 3.84, respectively (Table 3). 

Nine LR loci were mapped on seven chromosome arms (1BL, 2AL, 3AS, 3AL, 3BL, 6AS, and 6BL), and one major QTL was observed on chromosome arm 2AL (*QLR.ubo-2A.1*), with a log *p-*value of 4.08, and R^2^ of 4.92. As to SPAD, 21 QTLs were mapped on 12 chromosome arms (1AL, 2AL, 2BS, 3BS, 3BL, 4AL, 4BS, 5AL, 6BS, 6BL, 7AL, and 7BS), and three major QTLs were observed on chromosome arms 1AL (*QSPAD.ubo-1A.1*), 4BS (*QSPAD.ubo-4B.1*), and 5AL (*QSPAD.ubo-5A.1*), with a log *p-*value of 6.08, 6.61, and 6.81, and R^2^ of 7.87, 8.69, and 8.99, respectively. The global R^2^ of the multiple QTL model was 58.0% for ψs-c, 56.5% for ψs-s, 63.6% for OA, 25.7% for RWC-s, 44.1% for LR, and 50.2% for SPAD. On the basis of their concurrent allelic effects on OA and other related traits, eight major QTL clusters were identified: (i) *DR_QTL_cluster_1#* (RWC-s and ψs-s) on 1B, (ii) *DR_QTL_cluster_2#* (OA and ψs-c) on 2B, (iii) DR_QTL_cluster_3# (OA and RWC-s) on 4A, (iv) *DR_QTL_cluster_4#* (OA and SPAD) on 5A, (v) *DR_QTL_cluster_5#* (OA and RWC-s) on 6A, (vi) *DR_QTL_cluster_6#* (OA and RWC-s) on 6B, (vii) *DR_QTL_cluster_7#* (OA, RWC-s, and SPAD) on 6B, and (viii) *DR_QTL_cluster_8#* (OA and SPAD) on 7B (Table 4 and Figure 4).

These regions were selected for a more detailed analysis and comparative analysis with previously reported results on grain yield in both durum and bread wheat, as discussed hereafter. Durum wheat genes within the confidence intervals of the eight selected QTL hotspots were retrieved from EnsemblPlants database, together with their functional annotation (Appendix A). Gene Ontology (GO) enrichment analysis (Figure 5) and KEGG pathways reconstruction (Table 5) were used to further functionally characterize the genes included in the eight QTL clusters. In parallel, the bread wheat orthologous genes were identified, as well (see Appendix A), for the comparison of genes annotated in the syntenic regions of the two wheat species. KnetMiner [71] knowledge networks, constructed using bread wheat orthologs, were integrated to identify putative candidate gene(s) within the confidence interval of each QTL. The confidence interval of *DR_QTL_cluster_1#* on chromosome 1B corresponds to a physical interval of approximately 7.0 Mb with 46 high-confidence (HC) genes in the Svevo genome (Appendix A). Among the genes included in the interval, no GO terms were significantly enriched, while KEGG mapping annotated 21 genes to nine functional categories (Table 5). The two most notable candidates in the confidence interval are *TRITD1Bv1G127690*, which encodes a transmembrane protein with transporter activity homologous of the *Arabidopsis* Major facilitator superfamily MEE15, and *TRITD1Bv1G126800*, which encodes for a seven transmembrane MLO-like protein. The confidence interval of the *DR_QTL_cluster_2#* on chromosome 2B corresponds to a 3.2 Mb interval, which contains 63 high-confidence (HC) genes. GO terms associated to “stress response” and “antioxidant activity” were enriched among these genes (Figure 5), due to the presence of 10 peroxidase encoding genes in the QTL interval. KEGG mapping confirmed their annotation in secondary metabolism pathway (Table 5), acting in the phenylpropanoid biosynthesis. In addition to these peroxidase encoding genes that could act in drought-stress response and adaptation, the two most notable candidates in the interval are *TRITD2Bv1G263980*, encoding for a protein kinase and *TRITD2Bv1G264060*, which encodes a DDB1- and CUL4-associated factor-like protein 1. Both genes are located at the confidence interval boundaries and could be functionally related to OA-related aspects (Appendix A). Notably, the comparison of the syntenic physical region in *T. aestivum* Chinese Spring evidenced several gaps in the corresponding Svevo region: for 30 HC genes annotated in the Chinese Spring syntenic region, their Svevo orthologs are indeed included in unmapped scaffolds and could, thus, be included among the list of putative candidates. Even if none of these 30 missing genes are drought-stress responsive or OA-related functional annotation, the presence of these gaps in *T. durum* genome assembly and gene annotation could clearly impair candidate gene discovery. The confidence interval of *DR_QTL_cluster_3#* on chromosome 4A corresponds to approximately 3.5 Mb with 33 HC genes (Appendix A), mapped to seven KEGG functional categories (Table 5) and enriched in manganese transport-related GO terms (Figure 5). *TRITD4Av1G256080* and *TRITD4Av1G256120* indeed encode for membrane protein of ER body-like proteins, likely working as metal transporters. Additional genes annotated in the QTL interval include *TRITD4Av1G255460*, *TRITD4Av1G255480*, and *TRITD4Av1G255490*, encoding for three Glutathione S-transferases, *TRITD4Av1G255990*, encoding for an RNA-binding family protein with RRM/RBD/RNP motifs, and *TRITD4Av1G256200*, which encodes for a 5’-methylthioadenosine/S-adenosylhomocysteine nucleosidase. In addition, for this QTL hotspot, a gap between *T. durum* and *T. aestivum* chromosome assemblies was found, hence impairing a more proper identification of candidate genes. The confidence interval of *DR_QTL_cluster_4#* on chromosome 5A spans a physical interval of approximately 4.0 Mb, which contains 39 HC genes in the Svevo genome (Appendix A). Lyase activity was the unique GO term enriched among the genes within the QTL (Figure 5), while KEGG mapping assigned 17 of them to eight functional groups (Table 5). Interestingly, *TRITD5Av1G246840*, encoding for a putative Phospholipase D potentially implicated in multiple plant stress responses, and *TRITD5Av1G247330*, a putative Lipoxygenase required for jasmonic acid accumulation, were mapped to the “environmental information processing” category. In addition, *TRITD5Av1G247220*, encoding for an UDP-N-acetylglucosamine (UAA) transporter family protein, could be involved in the osmosensory signaling pathway and cell wall organization. The confidence interval of the *DR_QTL_cluster_5#* on chromosome 6A spans approximately 2.0 Mb, with 32 HC genes (Appendix A). While the “manganese binding” function was the only GO-enriched term (Figure 5), six of the genes included in the region encode for putative Cinnamoyl CoA reductases and were mapped to the “secondary metabolism”/“phenylpropanoid biosynthesis” pathways (Table 5). The two most notable candidates in the interval are *TRITD6Av1G217800* and *TRITD6Av1G218080*, which encode for two F-box protein PP2, and *TRITD6Av1G217670*, encoding for DREB1, a CRT-binding factor. The confidence interval of *DR_QTL_cluster_6#* on chromosome 6B spans a 5.9 Mb interval, which contains 40 HC genes in the Svevo genome (Appendix A). No GO term was found significantly enriched, while KEGG mapping annotated 20 genes to 12 different functional categories (Table 5). In particular, three genes encoding for MYB transcription factors potentially regulating different aspects of stress response were assigned to the “genetic information processing” category. Moreover, *TRITD6Bv1G133070*, orthologs of the CCT motif-containing response regulator protein coding gene of *Arabidopsis*, appears as an even more interesting candidate gene at this locus. The confidence interval of the *DR_QTL_cluster_7#,* located on chromosome 6B, spans approximately 1.2 Mb, with 14 HC genes, including *TRITD6Bv1G207930*, encoding for a protein kinase family protein/WD-40 repeat family protein 3. Both GO-enrichment and KEGG mapping did not identify other genes in this QTL interval (Table 5 and Figure 5). Finally, the confidence interval of *DR_QTL_cluster_8#* on chromosome 7B corresponds to a physical interval of approximately 1.5 Mb, which contains 20 HC genes in the Svevo genome, lacking GO enrichment or predominant KEGG annotations (Table 5 and Figure 5), but including *TRITD7Bv1G002000*, a gene encoding for a photosynthetic NDH subcomplex B3 (Appendix A).

## 4. Discussion

A number of authors have proposed OA as an important adaptive mechanism to support higher crop yield under stressful environmental conditions, as reviewed in [24,79]. Notably, grain yield differences have been shown to be positively correlated to OA in cereals [21,30,80,81,82,83], hence indicating OA as a valuable proxy to predict grain production [24]. This notwithstanding, the QTLome regulating OA in wheat and other crops remains basically unknown, the main reason being the difficulty to adequately screen the large number (>200) of (i) RILs of the mapping populations and/or (ii) accessions of GWAS mapping panels required for a meaningful QTL discovery. In field conditions, the collection of leaves and their processing must be completed rapidly to minimize the bias introduced by the time of sample collection in an adequately large number of genotypes, an essential prerequisite for identifying and accurately mapping QTLs [19,34]. The QTLome dissection of OA in cereals was first attempted in rice [37,38] and barley [39,40]. In bread wheat, Reference [84] mapped an osmoregulation gene locus [85] located on the short arm on chromosome 7A. However, OA and osmoregulation differ and have different functional meanings. While OA refers to a lowering of osmotic potential (ψs) due to an accumulation of osmolytes in response to a water deficit, osmoregulation refers to the ψs regulation by the addition/removal of osmolytes until the intracellular potential is approximately equal to that of the medium surrounding the cell [20]. The gene described by Morgan regulates turgor pressure and water content by osmotic adjustment [84,85], hence the term osmoregulation. In this study, OA was measured according to the “rehydration method” [23,86]. Although this method was criticized by [85], other authors consider it optimal for screening large populations [20,26,86,87,88,89] in view of its merits in terms of labor and cost-effectiveness as compared to the other methods [23]. In our experience, the rehydration of the leaf samples greatly facilitated (i) the cell sap extraction especially in samples collected in water-stressed plants and (ii) the OA screening of the 248 diverse accessions of the Durum Panel. Collectively, this resulted in high OA repeatability (h^2^) and a positive and negative correlation with RWC-s and LR, respectively. The positive correlation between OA and RWC clearly indicates an active physiological role of the former to maintain a more favorable water status of the plant, playing a key role for avoiding and mitigating the negative effects of water loss under drought. Overall, our results validate the effectiveness of the “Rehydration method” as an ideal option for handling the large number of samples required for the genetic dissection of the OA QTLome. 

### 4.1. GWAS Mapping and Comparative Analysis with Previous QTL Studies in Durum Wheat

Overall, eight major QTL hotspots were detected on the long arm of chromosomes, 1BL, 2BL, 4AL, 5AL, 6AL, 6BL, and 7BS. The use of flowering time (FT) as covariate for the GWAS analysis reduced the bias caused by the photoperiod-response (*Ppd*) locus and other loci that affect FT, hence allowing a more accurate estimate of QTL effects on a per se basis rather than due to effects related to variability in phenology. Notably, none of the eight major QTL hotspots evidenced by GWAS analysis overlapped with the osmoregulation gene locus described by Reference [85] in bread wheat. *DR_QTL_cluster_1#*, *DR_QTL_cluster_2#*, and *DR_QTL_cluster_5#* overlapped with Normalized Difference Vegetation Index (NDVI) loci identified in 2017 on the same Durum Panel under similar drought conditions using Unmanned Aerial Vehicles (UAV-Sequoia and UAV-Red-Edge), as well as ground-based platforms [60]. Additionally, *DR_QTL_cluster_3#* and *DR_QTL_cluster_5#* overlapped with chlorophyll content (SPAD) loci under drought described in Reference [60]. Both NDVI and SPAD have long been recognized for their ability to estimate crop biomass and predict grain yield [90,91,92,93,94]. *DR_QTL_cluster_2#* overlapped with grain yield, thousand-kernel weight, and NDVI loci previously reported in a durum wheat elite population tested in contrasting thermo-pluviometric conditions [76]. *DR_QTL_cluster_5#* co-mapped with *QRga.ubo-6A.2*, one of the most important loci for root growth angle in durum wheat [75], with thousand-grain weight, particularly under low water availability environments, as well as with grain yield, in the 183 elite accessions of the Durum Panel that were previously evaluated in 15 field trails under a wide range of water regimes [41]. *DR_QTL_cluster_3#* and *DR_QTL_cluster_7#* co-mapped with a major grain yield QTL reported by Reference [78] in an RIL population developed from the hexaploid wheat cross between Chinese Spring × SQ1 evaluated across a broad combination of 24 site × treatment × year combinations. The concurrent effects on grain-yield related traits reported herein fully support the conclusions of Reference [24] on OA being a valuable proxy with a positive effect on crop yield under water-limited conditions and not merely for survival under severe drought. These QTL hotspots will further enhance drought tolerance in durum wheat.

### 4.2. Candidate Genes 

By combining the physical confidence interval position of the QTL hotspots, functional prediction of annotated genes, and biological data mining, we investigated candidate genes from selected QTLs involved in OA and/or drought resistance. Putative drought candidate genes encoding for proteins involved in drought stress responses, as well as grain development, were mapped within the eight major selected QTL hotspots, even if some highlighted gaps in the Svevo genome assembly could hinder candidate gene identification at two QTL hotspots. Among the identified candidates, the seven transmembrane MLO-like protein (*TRITD1Bv1G126800*; *DR_QTL_cluster_1#*) was shown to act in drought and salt stress responses through signaling of the phytohormone abscisic acid (ABA) [95], with biological knowledge networks analysis strongly supporting its role in oxidative stresses, salt, and drought tolerance (Figure 6A). Similarly, the phospholipase D (PLD) coding gene (*DR_QTL_cluster_4#*) is also involved in ABA responses [96]. Interestingly, genes associated with manganese transport and binding were identified in both *DR_QTL_cluster_3#* and *DR_QTL_cluster_5#*. Moreover, exogenous application of Mn was recently shown to reduce the negative effects caused by drought, harsh temperature, and salinity, increasing ROS detoxification and secondary metabolism [97,98]. Despite the clear enrichment for stress response-related GO terms among genes at *DR_QTL_cluster_2#*, the identification of 10 tandemly duplicated peroxidase encoding genes acting in the phenylpropanoid biosynthetic make the construction of a knowledge network and putative candidate gene more complex and less reliable. Conversely, of considerable interest is, instead, the dehydration-responsive element-binding protein DREB (*DR_QTL_cluster_5#*; Figure 6B), that belongs to a family of plant-specific transcription factors that can specifically bind to DRE/CRT elements in the response to abiotic stresses, such as drought, salt, and low temperature [99,100], reviewed in [101]. In addition, *TRITD6Bv1G133070* (*DR_QTL_cluster_6#*), orthologs of the *Arabidopsis* CCT motif-containing response regulator protein, was shown to be involved in both the generation of circadian rhythms and long-term drought adaptation [102]. Finally, *TRITD7Bv1G002000* (*DR_QTL_cluster_8#*, affecting both OA and SPAD) encodes for chloroplast NAD(P)H dehydrogenase complex, involved in cyclic electron flow around photosystem I to produce ATP [103].

## 5. Conclusions

This study is the first to report QTLs for OA via GWAS mapping in wheat. From a methodological standpoint, our results support the validity of the “Rehydration method” as the fastest and most effective protocol for large-scale screening of OA under well-watered and drought conditions. The genetic variants within the Durum Panel evaluated herein allowed for the detection of significant loci for OA, ψs, RWC, LR, and SPAD, with eight multiple concurrent QTL hotspots, all unrelated to phenology, hence being more valuable from a breeding standpoint. Importantly, five of these clusters (*DR_QTL_cluster_1#*,*DR_QTL_cluster_2#*, *DR_QTL_cluster_3#*, *DR_QTL_cluster_5#*, and *DR_QTL _cluster _7#*) co-located with QTLs for grain yield and/or grain yield-related traits described in previous multi-environmental studies and, in one case (*DR_QTL_cluster_5#*), co-located with a major QTL controlling root growth angle which has been demonstrated to play a relevant role in maintaining the water access in deep soils during wheat terminal drought [104]. The candidate genes identified within the confidence intervals of selected drought response-specific QTL hotspots provide useful markers for future breeding schemes. Our results support the role of OA as an important drought-stress adaptive mechanism with beneficial effects on the plant water status in durum wheat.

## Figures and Tables

**Figure 1 genes-13-00293-f001:**
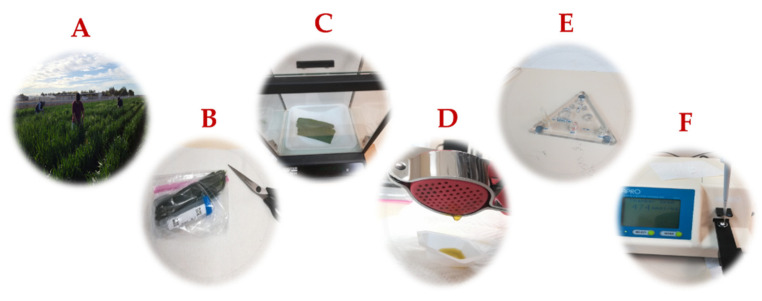
Major “Rehydration method” steps used at the Maricopa Agricultural Center (MAC) to measure osmotic adjustment (OA) and leaf relative water content (RWC): (**A**) Sampling of eight fully expanded homogeneous flag leaves before dawn. (**B**) Stacking the eight leaves and cutting off the tips. The remaining leaf parts (ca. 15 cm long) were cut in the middle to obtain two homogeneous pieces of similar weight, then mixed and inserted in Falcon 50 ml Conical Centrifuge Tubes. (**C**) Weighing of the leaf samples for RWC. (**D**) Collection of leaf cell sap for OA analysis using a garlic press. (**E**) Calibration of the osmometer (Wescor 5520) with sodium chloride solution of increasing concentration. (**F**) Pipetting ca. 10 µl of leaf cell sap onto a paper disc placed on the sampling cuvette of the osmometer (Wescor 5520).

**Figure 2 genes-13-00293-f002:**
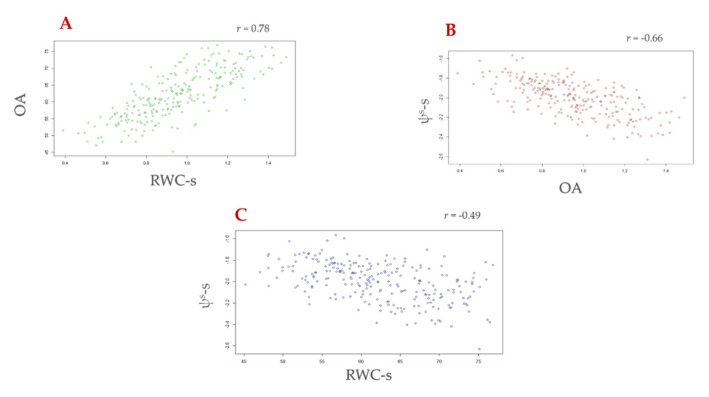
Scatter plot showing Pearson correlation between (**A**) osmotic adjustment (OA) and relative water content under drought (RWC-s), (**B**) OA and osmotic potential under drought (ψs-s), and (**C**) ψs-s and RWC-s. R-project [74].

**Figure 3 genes-13-00293-f003:**
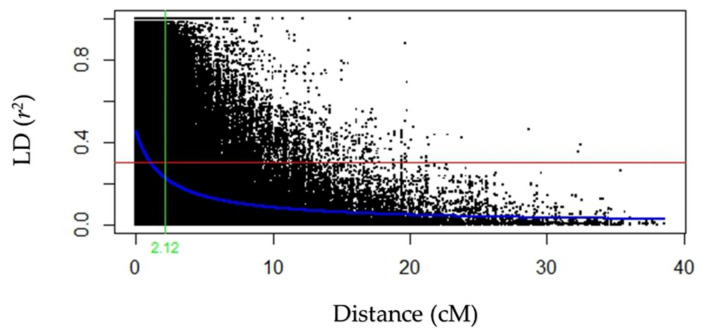
The rate of linkage disequilibrium (LD) decay of the 248 durum wheat elite accessions (Durum Panel). The Hill and Weir formula [56] was used to describe the LD decay of *r^2^*. The LD among SNPs in the Durum Panel was estimated using Haploview 4.2 [54]. The blue curve represents the model fit to LD decay (non-linear regression of *r^2^* on distance). A confidence interval of 2.12 cM for the QTLs is shown when LD (*r^2^*) is 0.3 (red line). R-project [74].

**Figure 4 genes-13-00293-f004:**
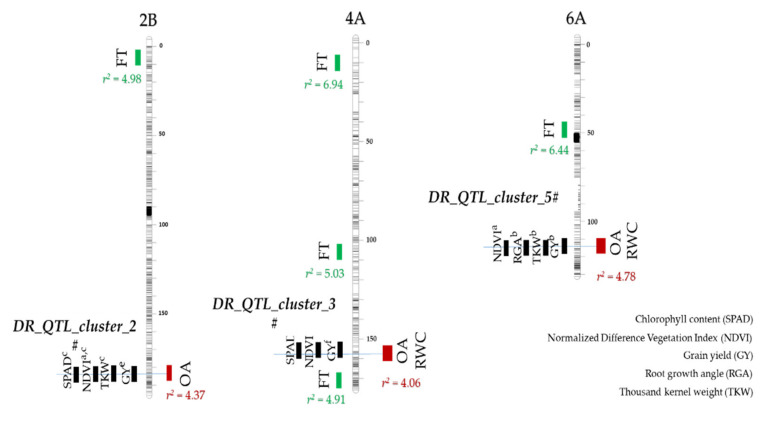
Chromosome position on the durum consensus map [54] and R^2^ of the QTL hotspots for osmotic adjustment (OA) and/or relative water content (RWC) on chromosome arms 2BL (*DR_QTL_cluster_2#*), 4AL (*DR_QTL_cluster_3#*), and 6AL (*DR_QTL_cluster_5#*), as well as their overlaps with QTLs previously reported in literature: ^a^ Reference [60], ^b^ Reference [75], ^c^ Reference [76], ^d^ Reference [41], ^e^ Reference [77], and ^f^ Reference [78]. The QTLs for flowering time (FT) are shown in green.

**Figure 5 genes-13-00293-f005:**
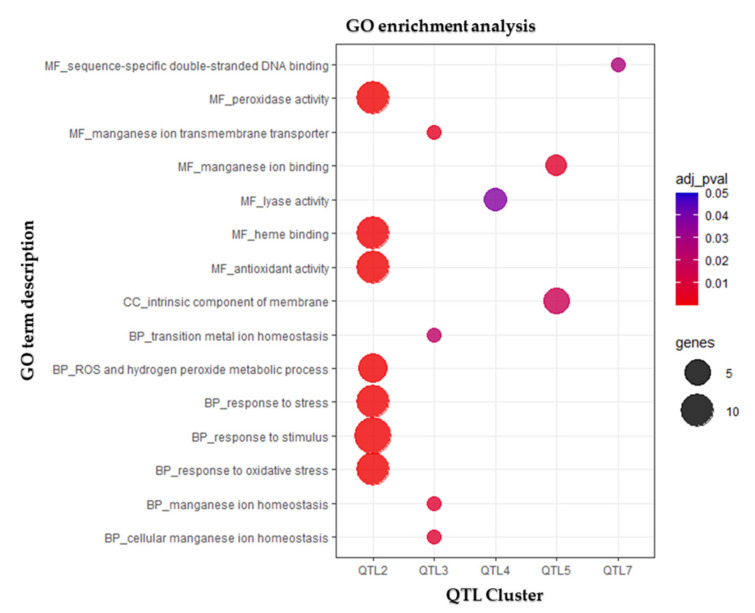
Gene Ontology (GO) enrichment analysis. Dot plot shows GO terms of biological processes (BP), molecular functions (MF) and cellular compartment (CC) identified using g:Profiler [68] to be enriched (adjusted *p-*value < 0.05) among the genes included in each QTL interval. The size of the dots is based on gene count enriched in the pathway, and the color of the dots represents the adjusted *p-*values.

**Figure 6 genes-13-00293-f006:**
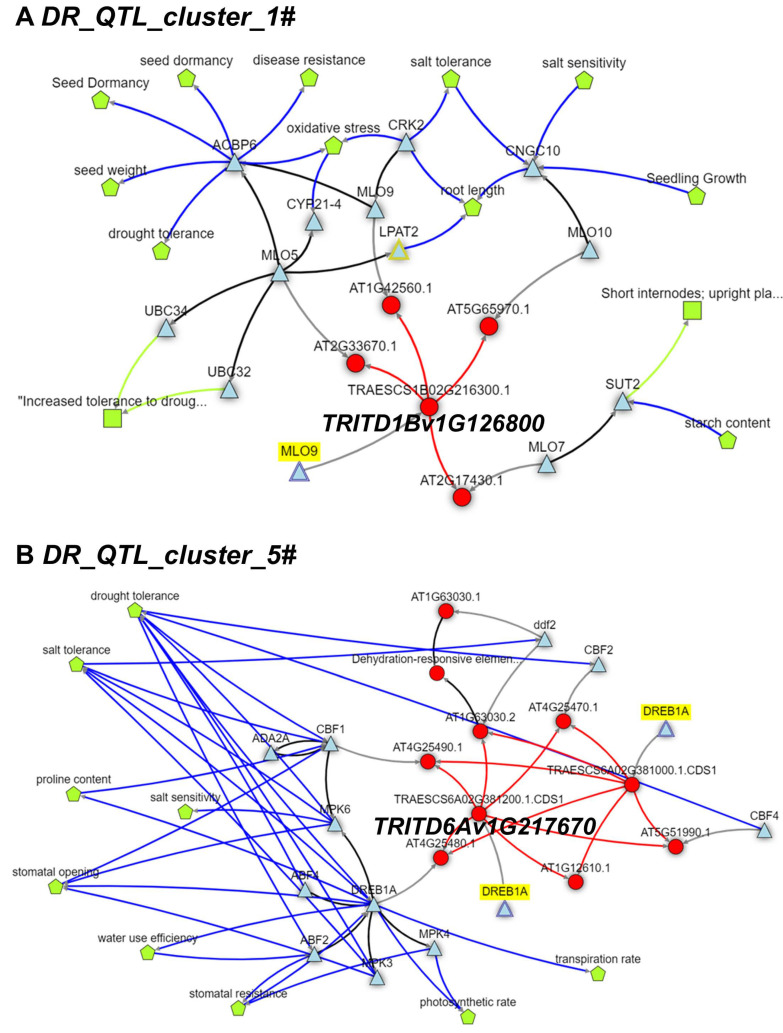
KnetMiner network views displaying knowledge networks of selected candidate genes Appendix A. *TRITD1Bv1G126800* (**A**), encoding for a seven transmembrane MLO-like protein (*DR_QTL_cluster_1#*), and *TRITD6Av1G217670* (**B**), which encodes for a dehydration-responsive element-binding protein DREB (*DR_QTL_cluster_5#*). Networks were constructed using the *Triticum aestivum* orthologous genes *TraesCS1B02G216300* and *TraesCS6A02G381200*, respectively, and can be accessed using the following links: https://knetminer.com/beta/knetspace/network/528cbd3a-52d5-40b5-91be-f59323db55a3 (accessed on 1 April 2021) and https://knetminer.com/beta/knetspace/network/44b31582-bfaa-495f-9272-87a4f06d40a6 (accessed on 1 April 2021).

**Table 1 genes-13-00293-t001:** Summary statistics and heritability (%) for osmotic adjustment (OA), osmotic potential under drought stress (ψs-s), osmotic potential under well-watered conditions (ψs-c), relative water content under drought stress (RWC-s), relative water content under well-watered conditions (RWC-c), leaf rolling (LR), and chlorophyll content (SPAD) in a panel of 248 durum wheat elite advanced lines and cultivars from worldwide.

Trait	Min.	Max.	Average	St. Dev.	*h^2^*
OA	0.38 ^a^	1.48	0.95	0.22	0.76
ψs-s	−2.63 ^a^	−1.56	−2.00	0.18	0.58
ψs-c	−1.45 ^a^	−0.75	−1.13	0.12	0.57
RWC-s	45.21 ^b^	76.88	62.10	7.11	0.78
RWC-c	89.9 ^b^	100.0	95.7	1.56	0.29
LR	2.86	9.60	6.13	1.52	0.84
SPAD	31.9	48.8	42.0	3.20	0.76

^a^ megapascal (MPa); ^b^ % RWC.

**Table 2 genes-13-00293-t002:** Pearson’s correlation plot among osmotic adjustment (OA), osmotic potential (ψ^s^) under full (--c) and deficit irrigation (--s), relative water content (RWC) under full (--c) and deficit irrigation (--s), leaf rolling (LR), and chlorophyll content (SPAD).

Trait	OA	ψ^s^-s	ψs-c	RWC-s	RWC-c	LR	SPAD
OA	1	−0.66 ***	0.33 ***	0.78 ***	0.11	−0.25 ***	0.04
ψ^s^-s	-	1	0.30 ***	−0.49 ***	−0.16 *	0.13 *	−0.06
ψ^s^-c	-	-	1	−0.08	0.02	−0.03	0.03
RWC-s	-	-	-	1	0.13 *	−0.30 ***	−0.02
RWC-c	-	-	-	-	1	−0.08	0.20 **
LR	-	-	-	-	-	1	−0.01
SPAD	-	-	-	-	-	-	1

*** *p-*value < 0.0001, ** 0.0001 < *p-*value < 0.001, * 0.001 < *p-*value < 0.01. *R-project* [74].

**Table 3 genes-13-00293-t003:** Significant GWAS-QTLs for osmotic adjustment (OA) and RWC-s (*p-*value < 0.001). QTL intervals were defined based on a confidence interval of ± 3.0 cM from the map position of the QTL tagging-SNPs. The rows with grey background indicate the QTLs affecting both OA and RWC-s. Position and peak marker of each QTL region are based on the tetraploid wheat consensus map [54].

	Osmotic Adjustment (OA)							
QTL	Marker	Chr.	Position (bp)	Position (cM)	Log *p-*Value	R^2^	Allele	Effect
*QOA.ubo-1A.1*	IWB27332	1A	508851821	88.3	3.07	3.10	C/T	−1.550
*QOA.ubo-1B.1*	IWB65251	1B	582533506	93.3	3.17	3.19	C/T	−0.097
*QOA.ubo-2A.1*	IWB34575	2A	36292525	46.6	3.11	3.11	A/G	0.128
*QOA.ubo-2A.2*	IWB39807	2A	768563743	206.8	3.08	3.31	C/T	0.090
*QOA.ubo-2B.1*	IWA2318	2B	656566640	133.0	3.89	4.07	C/T	−0.117
*QOA.ubo-2B.2*	wPt-0049	2B	781813758	185.8	4.13	4.37	A/T	0.141
*QOA.ubo-4A.1*	IWB38918	4A	644103100	139.7	3.11	3.12	A/G	−0.165
*QOA.ubo-4A.2*	IWB34029	4A	717060721	161.7	3.88	4.06	C/T	1.252
*QOA.ubo-4B.1*	IWB72203	4B	26616372	28.8	3.00	2.48	A/C	0.076
*QOA.ubo-5A.1*	IWB50381	5A	640718417	198.8	3.24	3.28	A/G	0.159
*QOA.ubo-6A.1*	wPt-2014	6A	505253000	91.2	4.01	4.23	A/T	0.162
*QOA.ubo-6A.2*	IWB70454	6A	596626025	117.1	4.45	4.78	C/T	0.181
*QOA.ubo-6B.1*	IWB33826	6B	437229717	75.3	3.12	3.13	A/G	−0.105
*QOA.ubo-6B.2*	IWB71722	6B	644758469	114.3	3.21	3.24	A/G	−0.086
*QOA.ubo-7B.1*	wPt-3147	7B	3571960	3.7	3.13	3.14	A/T	−0.095
	** Relative Water Content under water stress (RWC-s)**							
**QTL**	**Marker**	**Chr.**	**Position (bp)**	**Position (cM)**	**Log *p-*value**	**R^2^**	**Allele**	**Effect**
*QRWCs.ubo-1B.1*	IWB461	1B	628218198	45.3	3.70	3.24	C/T	−4.29
*QRWCs.ubo-2A.1*	IWB22184	2A	7224905	9.4	3.33	2.86	A/G	−4.25
*QRWCs.ubo-4A.1*	IWB66212	4A	687621664	140.7	3.02	2.53	A/C	2.74
*QRWCs.ubo-4A.2*	IWB56811	4A	697055522	147.2	4.83	3.95	C/T	−5.51
*QRWCs.ubo-4A.3*	IWB55093	4A	707177021	156.9	4.27	3.84	A/G	5.24
*QRWCs.ubo-4A.4*	IWA3449	4A	720085814	161.7	3.90	3.45	C/T	4.66
*QRWCs.ubo-6A.1*	IWA4603	6A	597277894	117.7	3.39	2.92	A/G	3.15
*QRWCs.ubo-6B.1*	IWA7962	6B	454884102	78.8	3.04	2.56	A/G	−6.92
*QRWCs.ubo-6B.2*	IWB71722	6B	644758469	114.3	3.00	2.44	A/G	−2.46

**Table 4 genes-13-00293-t004:** List of GWAS-QTL clusters identified in the Durum Panel and significantly associated with osmotic adjustment (OA), RWC under drought stress (RWC-s), osmotic potential under well-watered conditions (ψs-c) and drought stress (ψs-s), and leaf rolling (LR). The co-localization with previously known normalized difference vegetation index (NDVI), chlorophyll content (SPAD), root growth angle (RGA), thousand kernel weight (TKW), and grain yield (GY) QTLs is reported.

QTL Cluster	Chr.	Position (cM)	Trait	QTLs from Literature
*DR_QTL_cluster_1#*	1B	45.3	RWC-s, ψ^s^-s	UAV-Red-Edge-NDVI ^a^, TKW ^c^, GY ^d^
*DR_QTL_cluster_2#*	2B	185.3	OA, ψ^s^-c	Tractor-NDVI ^a^, TKW ^c^, NDVI ^c^, Chlorophyll content (SPAD) ^c^, GY ^e^,
*DR_QTL_cluster_3#*	4A	161.7	OA, RWC-s	Tractor-NDVI ^a^, UAV-Red-Edge NDVI ^a^, Chlorophyll content (SPAD) ^a^, GY^f^
*DR_QTL_cluster_4#*	5A	198.8	OA, SPAD	
*DR_QTL_cluster_5#*	6A	117.1	OA, RWC-s	UAV-Red-Edge-NDVI ^a^, RGA/TKW/GY ^b^,
*DR_QTL_cluster_6#*	6B	75.3	OA, RWC-s	
*DR_QTL_cluster_7#*	6B	114.3	OA, RWC-s	GY ^f^
*DR_QTL_cluster_8#*	7B	3.7	OA, SPAD	

^a^ Reference [60], ^b^ Reference [75], ^c^ Reference [76], ^d^ Reference [41], ^e^ Reference [77] and ^f^ Reference [78].

**Table 5 genes-13-00293-t005:** Summary of KEGG functional pathways mapped for genes included in each QTL interval, grouped based on metabolic activities.

Functional Category	QTL1	QTL2	QTL3	QTL4	QTL5	QTL6	QTL7	QTL8
Carbohydrate metabolism	2			1		1		
Energy metabolism						1	1	
Lipid metabolism	2		2	3		1		1
Nucleotide metabolism						1		
Amino acid metabolism	1	1	4	3	1		1	
Glycan biosynthesis and metabolism						1		
Metabolism of cofactors and vitamins				3				1
Biosynthesis of other secondary metabolites		10			6			
Genetic information processing	1		2			2	1	1
Environmental information processing	1	2	2	2		2	1	
Organismal systems	1					2		
Protein families: metabolism	5	2	1			1	1	
Protein families: genetic information processing	6	2		1	1	5		
Protein families: signaling and cellular processes	2	6	1	1	4	2	1	1
Unclassified: metabolism			1	3		1		
**KEGG mapped genes**	**21**	**23**	**13**	**17**	**12**	**20**	**6**	**4**
**Total genes**	**46**	**63**	**33**	**39**	**32**	**40**	**14**	**20**

## Data Availability

Not applicable.

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
