# Peer review of "Genome Wide Association Study Uncovers the QTLome for Osmotic Adjustment and Related Drought Adaptive Traits in Durum Wheat"

_genes, 2022, doi:10.3390/genes13020293_

Round 1
Reviewer 1 Report
OA is regarded as an important adaptive mechanism to support higher crop yield under stressful environmental conditions. The current study describes GWAS analysis of OA and RWC in a wide collection of durum wheat. OA in the durum panel was highly variable both in the fully irrigated plants and after 17 days of severe drought stress. The high variability of OA reflects the genetic diversity of the durum panel, allowing for the detection of significant loci of OA and related traits under drought. The variable durum panel showed also differences in all other measured traits including flowering time. The fact that 50% of the accessions were at the stage of flowering to grain filling during the stress period may be problematic since the water status of the plants is also dependent on the developmental state at the sampling day of OP. The authors used FT as a covariate for the GWAS analysis to reduce the effects caused due to variability in phenology. Although it might not solve all problems related to the differences in plant developmental stage at sampling dates, however, it reduces the problem.
The authors used the rehydration method although it is not an ideal method as mentioned in reference 23 and others. Nevertheless, the estimation of H2 was 0.57 and 0.58, for Ops and OPc, and the computed OA was 0.76 (Table 2) indicating high repeatability. The authors showed that this method was efficient for testing a large number of samples in a short time, which is required for meaningful QTL discovery.
QTL discovery by GWAS analysis is dependent on phenotyping and genotyping. The SNP genotyping and GWAS were well performed, allowing for further gene annotation analysis and interpretation of the results. The result of this study is interesting and need some minor modifications detailed below:
- Biomass measurements were not used in the analysis, therefore lines 117-119 and 152-162 can be deleted.
- Table 2 - I recommend adding the physical position of SNP, based on the Svevo genome.
- Please go over carefully to check references are correctly placed. For example: in Line 450 - References 21 and 88 do not describe “The gene”… osmotic adjustment”, and reference 51 is not active.
Author Response
We wish to thank the reviewers for their comments and suggestions.
We enclose the revised version of the manuscript that has been corrected accordingly to the reviewers’ comments, except for those in Fig. 4 on linkage disequilibrium as indicated below.
Please note that due to the inclusions of the revisions, the page number does not coincide with that of the original version.
Reviewer 1
Biomass measurements were not used in the analysis, therefore lines 117-119 and 152-162 can be deleted.
Response. We agree and have proceeded accordingly.
Table 2 - I recommend adding the physical position of SNP, based on the Svevo genome.
Response. As to the linkage disequilibrium decay rate reported in Figure 4, we have deliberately chosen to relate the marker-pairwise LD estimates (as r2) to the genetic distances in cM and not to the physical distances due to the well-known non homogeneous, sigmoidal distribution of recombination in the highly repetitive large genomes of the Triticeae.
Our SNP pairwise LD survey was carried out chromosome-wide and we sampled all marker pairs from 0 to 40 cM aside irrespectively if they were located in distal or in pericentromeric regions.
The presence of wide differences in the genetic-to-physical ratio observed for the distal versus pericentromeric chromosome regions due to suppression of recombination in pericentromeric regions has been first described by Paux E et al. (2008. A physical map of the 1-gigabase bread wheat chromosome 3B. Science. 2008 322(5898):101-4. doi: 10.1126/science.1161847) in the first detailed study of sequencing and assembly of chromosome 3B of bread wheat.
The same distribution has been confirmed in durum wheat using the durum consensus map of Maccaferri et al. 2015 (A high‐density, SNP‐based consensus map of tetraploid wheat as a bridge to integrate durum and bread wheat genomics and breeding. Plant biotechnology journal, 13(5), 648-663).
The sequencing and assembly of Svevo durum wheat cultivar allowed for a detailed chromosome-by-chromosome genetic-to-physical ratio survey in the durum wheat genome, reported in Supplemental materials paragraph 2.1.10. Genetic maps, marker projection on the durum wheat genome and genome-wide investigation of recombination rate in Maccaferri et al. (2019).
Figure 4 shows that LD r2 decays to 0.30 level (considered as a reference threshold to detect a marker-QTL association with high confidence) at 2.12 cM genome-wide, irrespectively of the chr. region considered (distal or pericentromeric).
To better detail the LD decay rate also in terms of physical distances, we report the physical distance to which the LD decays to r2 = 0.30 separately for distal and pericentromeric regions, respectively. These metrics were reported in Maccaferri et al. (2019, Nature Genetics) and for the modern cultivated durum 2.12 cM corresponded to: 2.12 cM x 1.79 Mb/cM = 3.79 Mb in distal regions, corresponding to a gene content of 49 High Confidence genes and to: 2.12 cM x 107.1 Mb/cM = 227.1 Mb in pericentromeric regions, corresponding to a gene content of 767 High Confidence genes.
Please go over carefully to check if the references are correctly placed. For example: in Line 450 - references 21 and 88 do not describe “The gene”…osmotic adjustment” and reference 51 is not active.
Response. Many thanks for noting these two mistakes that have now been corrected accordingly. Reference 51 is now active. Thanks for noting it.
Reviewer 2 Report
The manuscript presents high quality data on QTLs for osmotic adjustment and related traits in durum wheat GWAS. Data are very well analysed and presented. The subject of the studies are important for stability of durum wheat biomass and putatively yield. Although the paper is very well written, several questions needs to be possibly explained:
line 16: plants were watered until flowering? - it disagrees with description in methods (lines 88/89) – plants were watered till stage preceding heading.
lines 155-157 sentence "On April 5th ... genotypes." - redundant
line 288: Table S3 does not contain data on FT QTLs, and these data must be added to final version of the paper, some sheets in the file with Tables S3 should be removed.
Figure 3: Physical positions instead of genetic in determination of LD would be possibly better choice
Editorial:
line 22: GWA and later GWAS; PRCP in heading of Table S3 - abbreviations that possibly may need explanation first time
lines 37, 47, 48,146 - numbers of pages agrees with these in reference - no need to give them in text
Typos in headings to Figure S4 and S5 "structure"
line 261 some brackets redundant
Author Response
We wish to thank the reviewers for their comments and suggestions.
We enclose the revised version of the manuscript that has been corrected accordingly to the reviewers’ comments, except for those in Fig. 3 on linkage disequilibrium as indicated below.
Please note that due to the inclusions of the revisions, the page number does not coincide with that of the original version.
Reviewer 2
line 16: plants were watered until flowering? - it disagrees with description in methods (lines 88/89) - plants were watered till stage preceding heading.
Response. Many thanks for noting this mistake that has now been corrected accordingly.
lines 155-157 sentence “On April 5th…genotypes.” - redundant
Response. Many thanks for noting this mistake that has now been corrected accordingly.
line 288: Table S3 does not contain data on FT QTLs, and these data must be added to final version of the paper, some sheets in the file with Tables S3 should be removed.
Response. Many thanks for noting this mistake that has now been corrected accordingly. An additional Supplemental Table 4 now reports all the QTL that affected flowering time.None of them overlapped with QTL for osmotic adjustment.
Figure 3: Physical positions instead of genetic in determination of LD would be possible better choice.
Response. As to the linkage disequilibrium decay rate reported in Figure 4, we have deliberately chosen to relate the marker-pairwise LD estimates (as r2) to the genetic distances in cM and not to the physical distances due to the well-known non homogeneous, sigmoidal distribution of recombination in the highly repetitive large genomes of the Triticeae.
Our SNP pairwise LD survey was carried out chromosome-wide and we sampled all marker pairs from 0 to 40 cM aside irrespectively if they were located in distal or in pericentromeric regions.
The presence of wide differences in the genetic-to-physical ratio observed for the distal versus pericentromeric chromosome regions due to suppression of recombination in pericentromeric regions has been first described by Paux E et al. (2008. A physical map of the 1-gigabase bread wheat chromosome 3B. Science. 2008 322(5898):101-4. doi: 10.1126/science.1161847) in the first detailed study of sequencing and assembly of chromosome 3B of bread wheat.
The same distribution has been confirmed in durum wheat using the durum consensus map of Maccaferri et al. 2015 (A high‐density, SNP‐based consensus map of tetraploid wheat as a bridge to integrate durum and bread wheat genomics and breeding. Plant biotechnology journal, 13(5), 648-663).
The sequencing and assembly of Svevo durum wheat cultivar allowed for a detailed chromosome-by-chromosome genetic-to-physical ratio survey in the durum wheat genome, reported in Supplemental materials paragraph 2.1.10. Genetic maps, marker projection on the durum wheat genome and genome-wide investigation of recombination rate in Maccaferri et al. (2019).
Figure 4 shows that LD r2 decays to 0.30 level (considered as a reference threshold to detect a marker-QTL association with high confidence) at 2.12 cM genome-wide, irrespectively of the chr. region considered (distal or pericentromeric).
To better detail the LD decay rate also in terms of physical distances, we report the physical distance to which the LD decays to r2 = 0.30 separately for distal and pericentromeric regions, respectively. These metrics were reported in Maccaferri et al. (2019, Nature Genetics) and for the modern cultivated durum 2.12 cM corresponded to: 2.12 cM x 1.79 Mb/cM = 3.79 Mb in distal regions, corresponding to a gene content of 49 High Confidence genes and to: 2.12 cM x 107.1 Mb/cM = 227.1 Mb in pericentromeric regions, corresponding to a gene content of 767 High Confidence genes.
Editorial:
line 22: GWA and later GWAS;
Response: Many thanks for noting this inconsistency that has now been corrected accordingly. GWAS has been used throughout the manuscript.
PRCP in heading of Table S3 - abbreviations that possibly may need explanation first time
Response. Very sorry, I was unable to find this PRCP in heading of Table S3.
lines 37, 47, 48, 146: numbers of pages agree with these in references - no need to give them in text.
Response. Page numbers have now been eliminated from the references in main text.
typos in headings to Figure S4 and S5 “structure”
line 261: some brackets redundant
Response. Thanks for noting this. The revised text has been changed accordingly.
Thanks again for your comments and time. Should you have other questions, let us know.
Best regards,
Roberto Tuberosa and coauthors.